# Cardiac Remodeling and Arrhythmic Burden in Pre-Transplant Cirrhotic Patients: Pathophysiological Mechanisms and Management Strategies

**DOI:** 10.3390/biomedicines13040812

**Published:** 2025-03-28

**Authors:** Charilila-Loukia Ververeli, Yannis Dimitroglou, Stergios Soulaidopoulos, Evangelos Cholongitas, Constantina Aggeli, Konstantinos Tsioufis, Dimitris Tousoulis

**Affiliations:** 11st Department of Cardiology, Hippokrateio General Hospital, National and Kapodistrian University of Athens, 11527 Athens, Greece; harilila_med@hotmail.com (C.-L.V.); soulaidopoulos@hotmail.com (S.S.); dina.aggeli@gmail.com (C.A.); ktsioufis@hippocratio.gr (K.T.); drtousoulis@hotmail.com (D.T.); 21st Department of Internal Medicine, Laiko General Hospital, National and Kapodistrian University of Athens, 11527 Athens, Greece; cholongitas@yahoo.gr

**Keywords:** liver cirrhosis, cirrhotic cardiomyopathy, portal hypertension, left ventricular remodeling, QT prolongation

## Abstract

**Background**: Chronic liver disease (CLD) and cirrhosis contribute to approximately 2 million deaths annually, with primary causes including alcohol-related liver disease (ALD), metabolic dysfunction-associated steatotic liver disease (MASLD), and chronic hepatitis B and C infections. Among these, MASLD has emerged as a significant global health concern, closely linked to metabolic disorders and a leading cause of liver failure and transplantation. **Objective**: This review aims to highlight the interplay between cirrhosis and cardiac dysfunction, emphasizing the pathophysiology, diagnostic criteria, and management of cirrhotic cardiomyopathy (CCM). **Methods**: A comprehensive literature review was conducted to evaluate the hemodynamic and structural cardiac alterations in cirrhosis. **Results**: Cirrhosis leads to portal hypertension and systemic inflammation, contributing to CCM, which manifests as subclinical cardiac dysfunction, impaired contractility, and electrophysiological abnormalities. Structural changes, such as increased left ventricular mass, myocardial fibrosis, and ion channel dysfunction, further impair cardiac function. Vasodilation in the splanchnic circulation reduces peripheral resistance, triggering compensatory tachycardia, while the activation of the renin–angiotensin–aldosterone system (RAAS) promotes fluid retention and increases cardiac preload. Chronic inflammation and endotoxemia exacerbate myocardial dysfunction. The 2005 World Congress of Gastroenterology (WCG) and the 2019 Cirrhotic Cardiomyopathy Consortium (CCC) criteria provide updated diagnostic frameworks that incorporate global longitudinal strain (GLS) and tissue Doppler imaging (TDI). Prolonged QT intervals and arrhythmias are frequently observed. Managing heart failure in cirrhotic patients remains complex due to intolerance to afterload-reducing agents, and beta-blockers require careful use due to potential systemic hypotension. The interaction between CCM and major interventions, such as transjugular intrahepatic portosystemic shunt (TIPS) and orthotopic liver transplantation (OLT), highlights the critical need for thorough preoperative cardiac evaluation and vigilant postoperative monitoring. **Conclusions**: CCM is a frequently underdiagnosed yet significant complication of cirrhosis, impacting prognosis, particularly post-liver transplantation. Early identification using echocardiography and thorough evaluations of arrhythmia risk in cirrhotic patients are critical for optimizing management strategies. Future research should focus on targeted therapeutic approaches to mitigate the cardiac burden in cirrhotic patients and improve clinical outcomes.

## 1. Introduction

Chronic liver disease (CLD) and cirrhosis contribute to two millions deaths annually across the globe. Alcohol-related liver disease (ALD), metabolic dysfunction-associated steatotic liver disease (MASLD, previously referred to as nonalcoholic fatty liver disease (NAFLD)), and chronic hepatitis B and C infections are the main causes of chronic liver disease and cirrhosis [1,2,3,4].

The increasing incidence of cirrhosis in European countries poses a potential health challenge. MASLD, in particular, has emerged as an important concern for CLD not just in Europe but worldwide, with an estimated global prevalence of 30% [5]. The term MASLD replaced NAFLD to better represent its link to metabolic dysfunction in 2020 [3,6,7]. The characteristics of this condition are insulin resistance and liver fat deposition, accompanied by persistent low-grade inflammation, oxidative stress, mitochondrial dysfunction, and gut microbiota imbalance which can result in long-term cardiovascular and liver-related complications. Furthermore, metabolic dysfunction-associated steatohepatitis (MASH), which is a subtype of MASLD and characterized by inflammation and liver cell damage, has become the most rapidly increasing cause of chronic liver failure requiring liver transplantation [8,9].

Liver cirrhosis is the outcome of CLD, with disease progression mainly driven by related complications. The main mechanism behind the cirrhosis-associated complications is portal hypertension, which leads to intestinal congestion and the translocation of bacteria or endotoxins into the systemic circulation, leading to a subclinical, persistent inflammatory state. This condition, combined with liver insufficiency and disrupted synthesis or metabolism of various substances (proteins and lipids/lipoproteins), is linked to cardiac dysfunction in patients with cirrhosis. Individuals with end-stage liver disease face a fivefold increased risk of heart failure, particularly heart failure with preserved ejection fraction (HFpEF), a subject of significant interest in cardiology [10,11].

Cirrhotic cardiomyopathy (CCM) is a condition frequently observed in patients with end-stage liver disease and is defined as subclinical cardiac dysfunction, marked by reduced cardiac contractility during physical activity and/or impaired diastolic relaxation with electrophysiological irregularities in patients without pre-existing heart conditions [11]. Cardiac dysfunction in patients with cirrhosis is often concealed due to the reduced peripheral resistance caused by arterial vasodilation in the splanchnic circulation, which decreases the left ventricular afterload and overall cardiac workload [12]. However, during acute events, such as infections and gastrointestinal bleeding, or following a transjugular intrahepatic portosystemic shunt (TIPS) procedure, the diminished responsiveness to adrenergic stimulation can reveal underlying cardiac dysfunction, potentially leading to heart failure [13,14,15]. The main issue of this entity is its link with cardiovascular disease following liver transplantation, with 7–24% of early post-transplant deaths imputed to heart failure.

Additionally, studies have shown that cardiac decompensation is the leading cause of death in patients with cirrhosis undergoing a TIPS procedure, with 20% developing acute heart failure within a year of TIPS placement [16,17]. However, in the pre-transplantation cirrhotic patients, the status of the left ventricle and the presence of arrhythmias are not well documented. Therefore, in this review, we aimed to assess left ventricular remodeling and the presence of arrhythmias in pre-transplantation cirrhotic patients [18].

## 2. Epidemiology of Cirrhotic Cardiomyopathy

Since CCM is characterized by subclinical cardiac dysfunction, it becomes particularly significant when stressors such as fluid overload disrupt hemodynamics. In the decompensated phase of cirrhosis or in advanced stages where inflammation becomes the dominant pathophysiological process, CCM can be most frequently observed. However, CCM often remains asymptomatic, and many cases likely go underdiagnosed, resulting in an underestimate of its prevalence and incidence. Additionally, the existence of different diagnostic criteria contributes to the higher variation in the prevalence of CCM. As a result, references indicate a prevalence range of 50–70% based on the 2005 Cirrhotic Cardiomyopathy Consortium (CCC) criteria and 29–55.7% based on the updated 2019 criteria [19,20,21,22].

A comprehensive diagnostic approach that incorporates various clinical echocardiographic findings and biomarker assessments is necessary due to the distinct pathophysiology of CCM. The initial definition of CCM was established in 2005 at the World Congress of Gastroenterology (WCG) in Montreal and was revised in 2019 to reflect advancements in ultrasound diagnostics [23].

## 3. Pathophysiology

### 3.1. Systemic Hemodynamic Changes in Cirrhotic Patients

Clinically significant portal hypertension is a defining feature of decompensated cirrhosis and remains an indicator of poor prognosis. According to Ohm’s law, the change in pressure across the portal system (ΔP) is directly proportional to blood flow (Q) and vascular resistance (R), as represented by the equation ΔP = Q × R [24,25]. Severe liver fibrosis increases resistance to blood flow. This heightened resistance triggers the release of various vasoactive substances, such as adrenomedullin, calcitonin, carbon monoxide, endocannabinoids, and nitric oxide, while the decompensated liver’s impaired metabolic capacity reduces their breakdown [26,27].

Porto-systemic collaterals form diverting and segmenting portal blood flow as angiogenesis progresses. The combination of increased blood flow and resistance elevates portal pressure, leading to complications like ascites, hydrothorax, and variceal bleeding [28,29]. Furthermore, vasoactive mediators, mainly nitric oxide, exert vasodilatory effects on systemic circulation, reducing venous return and causing a compensatory increase in baseline heart rate to sustain cardiac output. This leads to a hyperdynamic circulatory state typical in cirrhosis [30].

However, while blood pools in the splanchnic circulation, the rest of the systemic circulatory system experiences relative arterial hypotension. Peripheral vasoconstriction receptors become downregulated, and the renin–angiotensin–aldosterone system (RAAS) is activated in an effort to enhance organ perfusion [31]. The cardiovascular system’s capacity to further increase cardiac output becomes limited under additional stressors, such as physical activity, emotional stress, or sepsis. In patients with pre-existing cardiovascular disease, these hemodynamic changes associated with cirrhotic cardiomyopathy can contribute to the risk of multi-organ failure [32] (Figure 1).

Microvascular disturbances play a pivotal role in the progression of liver cirrhosis and its complications. While intrahepatic microvascular dysfunction is well documented, the impact of extrahepatic microcirculatory alterations on multi-organ failure in end-stage cirrhosis remains less understood [33]. Studies assessing sublingual microcirculation in cirrhotic patients, both with and without sepsis, have demonstrated significant reductions in microvascular perfusion in decompensated cirrhosis, primarily due to vasoconstriction [34]. Increased post-occlusive hyperemia—a hallmark of microvascular dysfunction—and diminished peripheral microcirculatory flow are prevalent in advanced disease stages. These abnormalities are suggestive of precapillary shunt formation, resembling microvascular derangements seen in sepsis, which are associated with poorer clinical outcomes and a heightened risk of multi-organ failure. A clear correlation between microcirculatory impairment, cirrhosis severity, and organ dysfunction underscores its potential role as a therapeutic target in advanced liver disease [35].

Compelling experimental evidence also supports the association between left ventricular contractility and tumor necrosis factor-alpha (TNFα)-induced activation of the NF-κB-iNOS signaling pathway, oxidative injury, and disrupted β-receptor signaling. Furthermore, data in humans demonstrate a correlation between the severity of altered cardiovascular dynamics and systemic inflammation, as assessed by the measurement of circulating IL-6, IL-8, and soluble IL-33 receptor levels. Notably, elevated IL-6 levels independently predicted the incidence of fatal acute-on-chronic liver failure (ACLF) over a median follow-up period of three years [36,37,38].

### 3.2. Changes at the Myocardial Level

At the cellular level, myocardial changes in cirrhosis impact both systolic and diastolic function. Myosin heavy chains interact with thin actin filaments to form cross bridges during active cardiac contraction (systole). Myosin is present and is more effective in alpha subunits and less effective in beta subunits. In CCM, there is an upregulation of the weaker beta myosin subunit, impairing contractile strength. Similar diastolic abnormalities are observed in other types of cardiomyopathy as well [39,40].

Histologically, CCM is associated with myocardial hypertrophy and subendothelial edema, which can progress to fibrosis. The early stage of myocardial fibrosis is diffuse and potentially reversible, while more advanced subendothelial segmental fibrosis is considered permanent [41,42]. The extent of fibrosis can be evaluated using magnetic resonance imaging (MRI), with permanent fibrosis detected through late gadolinium enhancement techniques.

Other myocardial changes associated with cirrhosis include increased left ventricular mass, enlarged LV end-diastolic volumes, and greater left atrium volumes. These changes reflect the structural and functional adaptations of the heart to chronic hemodynamic stress imposed by cirrhosis and the hyperdynamic circulatory state [43,44].

### 3.3. Causes That Lead to Cardiac Dysfunction in Cirrhotic Patients

1. MASLD is defined by the presence of hepatic steatosis in conjunction with diabetes, obesity, or other indicators of metabolic dysfunction. Its incidence has been rising steadily, with prevalence rates reaching up to 30% in developed nations [9,45,46]. In the U.S, MASLD has become one of the leading cause of cirrhosis among liver transplant (LT) candidates, and a similar trend has been observed by the Australia and New Zealand Liver Transplant Registry (ANZLTR) [47,48]. Consequently, we should be aware of the reclassification of cases that were once labeled as cryptogenic cirrhosis, underscoring the need for more targeted liver therapies. Lifestyle modifications and weigh loss remain the only proven and effective treatment for MASLD [49].

A recent systematic review [46] has shown a significant link between MAFLD and coronary artery disease (CAD), both fatal and non-fatal, with the risk of CAD increasing as liver fibrosis advances, independent of other cardiovascular risk factors. In LT candidates with decompensated cirrhosis, metabolic syndrome features that support a diagnosis of MASLD can be challenging to detect due to a dysregulated lipid profile, a chronic catabolic state masking insulin resistance, obesity (vs ascites), and persistent hypotension from the chronic vasodilatory state [46].

MASLD has also been implicated as an important contributor to cardiac remodeling, which can predispose to electrical abnormalities, diastolic dysfunction, and heart failure. A large retrospective analysis involving nearly one million patients found that those with MASLD had a notably higher risk of developing heart failure, independent of other risk factors [50]. The pathophysiological mechanisms connecting MASLD to progressive myocardial dysfunction are not fully understood, but proposed factors include insulin resistance, mitochondrial dysfunction, RAAS activation, systemic inflammation, and gut microbiota imbalances [49]. More specifically, the pathophysiological substrate includes the following:

Insulin Resistance and Metabolic Dysregulation: Insulin resistance, a hallmark of MASLD, contributes to dyslipidemia, hypertension, and impaired glucose metabolism. These metabolic disturbances are well-established risk factors for the development and progression of cardiovascular diseases [51].

Systemic Inflammation: Chronic low-grade inflammation is prevalent in MASLD and plays a crucial role in cardiovascular pathology. Elevated levels of pro-inflammatory cytokines, such as TNF-α and interleukins IL-6 and IL-1β, have been observed. These cytokines can lead to endothelial dysfunction, a precursor to atherosclerosis and other cardiovascular conditions [52].

Atherogenic Dyslipidemia: MASLD is often associated with an unfavorable lipid profile, characterized by increased triglycerides and decreased high-density lipoprotein cholesterol. This lipid imbalance promotes the development of atherosclerotic plaques, thereby elevating the risk of cardiovascular events [51].

Endothelial Dysfunction: The endothelium plays a vital role in vascular health, and its dysfunction is a key event in the progression of cardiovascular diseases. In MASLD, factors such as oxidative stress and inflammatory cytokines impair endothelial function, leading to reduced nitric oxide availability and compromised vasodilation [52].

The Gut–Liver Axis and Microbiota Alterations: Emerging research suggests that alterations in the gut microbiota contribute to the pathogenesis of MASLD and its cardiovascular implications. Dysbiosis may lead to increased intestinal permeability, allowing endotoxins to enter the portal circulation, thereby exacerbating hepatic inflammation and influencing the systemic cardiovascular risk [53].

Genetic and Epigenetic Factors: Individual genetic predispositions and epigenetic modifications can influence the susceptibility to both MASLD and cardiovascular diseases. These factors may affect lipid metabolism, inflammatory responses, and vascular function, thereby modulating disease progression and the cardiovascular risk [52].

Understanding these interconnected mechanisms is essential for developing targeted therapeutic strategies aimed at reducing the cardiovascular risk in individuals with MASLD. The early detection and management of metabolic disturbances, along with interventions addressing inflammation and endothelial dysfunction, are crucial in mitigating the cardiovascular burden associated with MASLD.

2. Alcohol-induced dilated cardiomyopathy can occur even in the absence of cirrhosis. However, with alcohol use disorder affecting up to a million individuals, the true prevalence of this condition is likely underestimated and underdiagnosed. Alcoholic cardiomyopathy can be distinguished from CCM by factors such as a recent history of excessive alcohol intake, pronounced ventricular dilation, reduced ejection fraction (EF) during echocardiography, and more overt heart failure symptoms. Encouragingly, cardiac function and EF often improve with alcohol abstinence. Evidence supporting standard heart failure treatments for alcohol-related cardiomyopathy is limited. Since LT candidates are traditionally required to maintain at least three to six months of abstinence before being considered for transplant, alcohol cardiomyopathy rarely poses an ongoing problem in prospective LT patients [54,55,56].

3. Cirrhosis was once thought to offer some protection against CAD due to factors such as lower cholesterol levels, thrombocytopenia, coagulopathy, and a hyperdynamic circulatory state [57]. However, a recent metanalysis found the prevalence of obstructive CAD in cirrhotic patients to be approximately 12.6%, comparable to that of the general population. This indicates that patients with cirrhosis are still susceptible to CAD-related complications, including ischemic cardiomyopathy and arrhythmias [58].

Additionally, both platelet count and INR have proven to be unreliable indicators of coagulation status in cirrhosis, as a complex rebalancing of thrombotic factors actually heightens both bleeding and clotting risks. Proactive management of CAD risk factors, such as the use of statins to address dyslipidemia, is recommended, especially considering the challenges of revascularization in this high-risk group. Given that CAD is increasingly recognized as a contributor to cardiac dysfunction in cirrhosis, comprehensive cardiovascular risk assessments before liver transplantation are essential. Patients with multiple risk factors or metabolic syndrome may benefit from coronary angiography and possible revascularization prior to the transplant [59,60].

4. Both light chain and transthyretin amyloidosis (including hereditary and non-hereditary types) are known to cause restrictive cardiomyopathy. The early detection of amyloidosis is challenging due to subtle initial symptoms and often unremarkable transthoracic echocardiograms (TTEs). The prognosis for these patients remains poor, with survival rates of up to five years despite chemotherapy treatments for light chain amyloidosis. Liver transplantation may be considered for hereditary transthyretin amyloidosis, as the liver is the source of the mutant protein and LT can halt further production of the defective protein. However, outcomes remain limited as cardiomyopathy can progress even after LT [61].

5. Haemochromatosis, a common genetic condition, affects up to 30% of the Australian population as carriers of a mutation associated with hereditary heamochromatosis. Cardiomyopathy is a serious, irreversible consequence of untreated iron overload, occurring in both acquired and hereditary cases. Haemochromatosis significantly raises the cardiovascular mortality risk up to 14-fold, with this elevated risk persisting after liver transplantation. Cases of heart failure after transplant for hereditary heamochromatosis have been documented, with successful management achieved through regular venesection and heart failure-specific treatments [62,63,64].

### 3.4. The Role of Inflammation in Cirrhosis

The hyperdynamic circulatory state has traditionally been viewed as the primary driver of extrahepatic organ dysfunction in advanced liver dysfunction, leading to conditions such as cirrhotic cardiomyopathy, hepatorenal syndrome, and hepatopulmonary syndrome [65]. Recent insights indicate that the impact on other organs involves more than just hemodynamic changes. For instance, the strong production of vasodilators such as nitric oxide (NO), carbon monoxide (CO), endocannabinoids, and prostacyclin was previously difficult to fully interpret. Patients with advanced liver cirrhosis often exhibit subclinical inflammation that is driven by a variety of circulating cytokines. Mesenteric congestion from portal hypertension promotes bacterial translocation and endotoxemia, as increased pressure weakens the intestinal mucosal barrier and alters the gut microbiome. The disruption allows bacteria and pathogen-associated molecular patterns to enter the bloodstream via the mesenteric lymph nodes. The breakdown of cell–cell connections and the resulting edema in the intestinal wall due to venous congestion are thought to facilitate this bacterial translocation. While the intestinal immune system generally neutralizes translocated bacteria to prevent infection, a chronic inflammatory response still ensues. As liver cirrhosis progresses, this inflammation intensifies, with the released cytokines eventually affecting multiple organs, including the heart [66,67,68,69].

## 4. Discussion

For a diagnosis of CCM, the absence of chronic underlying heart disease is essential. The initial diagnostic criteria for CCM were introduced in 2005 at the WCG in Montreal, and they are divided into three primary categories. The first two categories focused on evaluating systolic and diastolic function, while the third category included supportive indicators like electrophysiological or structural heart changes and lab markers such as elevated brain natriuretic peptide (BNP) or N-terminal pro-BNP levels [11,70]. With significant advancements in transthoracic echocardiography, particularly in tissue Doppler imaging (TDI) and myocardial strain analysis, new echocardiographic measures have been developed for assessing systolic and diastolic function. Additionally, in 2015, the American Society of Echocardiography (ASE) and the European Association of Cardiovascular Imaging (EACVI) recommended including myocardial strain analysis, specifically global longitudinal strain (GLS), alongside left ventricular ejection fraction (LVEF) to evaluate left ventricular contractility. Furthermore, in 2016, they revised the guidelines for diagnosing and classifying left ventricular diastolic dysfunction [71,72].

In 2019, a multidisciplinary international group, the CCC, proposed updated criteria for CCM. The new criteria, based on the unique pathophysiology of CCM which involves a hyperdynamic circulation with low peripheral resistance and increased venous return, incorporate GLS and tissue Doppler velocity, providing a more precise assessment of systolic and diastolic function [21,73] (Table 1).

### 4.1. The Role of Systolic Dysfunction

Systolic dysfunction is a key criterion for CCM, caused by impaired myocardial contractility. The 2005 WCG defined systolic dysfunction as an LVEF below 55% at rest and/or a diminished ability to increase contractility during physical activity [73,74].

In 1953, Kowalski and Abelmann were the first to report that cirrhosis is linked to a hyperdynamic circulatory state, defined by elevated cardiac output and reduced peripheral vascular resistance [12]. These observations have since been validated by numerous subsequent studies. Due to a hyperdynamic circulatory state, most CCM patients exhibit normal or enhanced left ventricular function at rest. This state, characterized by elevated stroke volume, increased preload, a faster heart rate, and reduced afterload, often masks the heart’s underlying contractile impairment. Consequently, systolic dysfunction becomes evident primarily under stress conditions. It is believed that individuals with CCM have difficulty increasing their LVEF and cardiac output during stress [74,75]. This theory, initially proposed in 1969 in research on alcoholic cirrhosis, has since been corroborated by additional studies. For instance, Wong et al. reported in 2001 that patients with liver cirrhosis, regardless of its origin, were unable to enhance LVEF or cardiac output during exercise. This phenomenon is attributed to the heart already operating near its maximal capacity at rest, leaving little reserve for increased demand [76].

Evidence indicating a post-transplant decline in LVEF, attributed to a rapid increase in afterload and the resolution of hyperdynamic circulation, suggests an underlying contractile dysfunction inherent to CCM. Shin et al. further highlighted altered cardiac mechanics in CCM, demonstrating a rightward shift in the cardiac pressure–volume curve. This shift, along with decreased end-systolic elastance and arterial elastance, reflects reduced ventricular contractility and diminished arterial load integration in CCM [77].

The molecular mechanisms contributing to these contractile abnormalities include elevated levels of cardiac depressant substances, disrupted β-adrenergic signaling, and chronic sympathetic nervous system (SNS) activation. In cirrhosis, splanchnic and systemic vasodilation lead to central hypovolemia, triggering SNS activation to maintain peripheral perfusion [78,79,80]. Prolonged SNS activation results in high catecholamine levels, damaging cardiac cells, downregulating β-adrenergic receptors, and desensitizing them by uncoupling G-proteins, leading to decreased cyclic adenosine monophosphate (cAMP) production [77].

Additionally, low-grade inflammation, driven by endotoxemia and elevated cytokines, exacerbates the production of vasoactive mediators like NO, CO, and endothelin. These mediators disrupt myocardial calcium channel function, causing intracellular calcium imbalance, which impairs myocardial contraction and relaxation. Studies in animal models suggest that NO and CO increase cyclic guanosine monophosphate (cGMP), reducing cAMP and exerting negative inotropic effects. Preclinical research has also shown that pro-inflammatory cytokines such as tumor necrosis factor (TNF)-α, interleukin (IL)-6, and IL-1β are significantly elevated in cirrhotic hearts. Blocking these cytokines has been shown to reduce inflammation, cardiac remodeling, and contractile dysfunction, offering potential therapeutic pathways [75,78,79].

GLS analysis using speckle-tracking echocardiography (STE) has proven effective in detecting subclinical myocardial dysfunction across various conditions. GLS is particularly valuable for identifying myocardial contractile dysfunction in patients with preserved LVEF, as longitudinal contractile function typically deteriorates before radial contractile function [73].

GLS is measured using specialized software and exhibits lower intra- and inter-observer variability compared to LVEF. This makes GLS a sensitive marker for detecting systolic dysfunction [81]. Unlike Doppler methods, GLS is not angle-dependent. However, it is influenced by factors such as load, age, and gender. Despite these influences, GLS has become a key parameter for assessing cardiac performance, as it is capable of identifying subclinical systolic dysfunction in various conditions. In addition, GLS is correlated with the prognosis of cirrhotic patients in intensive care, even when conventional echocardiography does not reveal any apparent dysfunction [82,83]. This suggests that GLS can provide valuable insights into myocardial function that may not be captured by traditional echocardiographic measures [84].

A recent meta-analysis revealed that patients with cirrhosis exhibit lower GLS compared to controls. Furthermore, the severity of cirrhosis was significantly associated with greater GLS reduction. Future research should focus on detailed analyses to clarify the differences in GLS among cirrhotic patients based on the severity of their disease [85].

Numerous studies employing speckle-tracking analysis in cirrhotic patients have been conducted, though many have small sample sizes and lack robust subgroup analyses or prognostic data. The relationship between GLS and liver disease severity has been examined using both the Child–Pugh classification (compensated cirrhosis in class A, intermediate severity in class B, and advanced decompensated disease in class C) and the model for end-stage liver disease (MELD) score, which is strongly correlated with liver disease prognosis [86,87,88].

Findings on the association between GLS and cirrhosis severity have been inconsistent. While some studies found no significant difference in GLS values across Child–Pugh classes or MELD scores [89,90], others reported a significant association, with increased absolute GLS values in patients with more severe cirrhosis [22,91]. Additionally, portal hypertension appears to influence GLS, with patients exhibiting signs of portal hypertension, thereby showing increased GLS values similar to those with decompensated cirrhosis [92].

The prognostic value of GLS in cirrhosis remains debated. Some studies reported no association between GLS and adverse outcomes in hospitalized patients or outpatients with cirrhosis [93,94], while others suggested that increased absolute GLS might correlate with advanced cirrhosis and a poor prognosis [95]. However, low absolute GLS values were associated with worse outcomes in specific patient groups, such as those undergoing TIPS procedures [96].

These conflicting findings suggest that while elevated GLS may reflect disease severity in advanced cirrhosis, markedly reduced GLS values might be indicative of a poor prognosis, even in patients with compensated liver disease.

Studies have reported no significant differences in LVEF across Child–Pugh classes or MELD scores, suggesting that overt systolic dysfunction at rest is uncommon in cirrhosis.

In most studies, LV-GLS was assessed offline using speckle-tracking echocardiography. It was calculated as the average peak longitudinal strain across 17 segments of the left ventricle (LV), obtained from the apical 4-chamber, 3-chamber, and 2-chamber views [81,84,85].

At the same time, changes in geometric patterns of the LV and LA are common and independently linked to all-cause mortality and CCM. These markers hold promise for risk stratification and redefining CCM [97].

Furthermore, cardiac morphological alterations, such as left atrial enlargement, have been associated with the severity of liver cirrhosis. Studies have shown that left atrial size positively correlates with both Child–Pugh and MELD scores, suggesting that atrial remodeling progresses with worsening liver function [98].

### 4.2. The Role of Diastolic Dysfunction

Diastolic dysfunction is characterized by reduced ventricular compliance, resulting in impaired relaxation and filling. The most common histopathological changes in the ventricles include myocardial hypertrophy, increased subendothelial edema, and fibrosis [11,99]. These changes are primarily driven by the overactivation of the neurohormonal systems (SNS and RAAS), mechanical strain, and the inflammatory pathway typical of advanced cirrhosis with a hyperdynamic circulatory state [75].

Inflammatory mediators, such as IL-6, IL-8, IL-1β, TNF-α, and transforming growth factor-beta (TGF-β), activate stress response pathways in the myocardium, leading to cardiomyocyte apoptosis and necrosis. Angiotensin II (AT II), a key component of the RAAS system, contributes to diastolic dysfunction by promoting fluid retention, volume overload, and myocardial remodeling. Animal studies reveal that AT II stimulates the production of extracellular matrix proteins and upregulates TGF-β through its AT1 receptor [75,78].

Cardiac fibrosis, a hallmark of pathological hypertrophy and heart failure, is marked by an accumulation of collagen and other extracellular matrix components in the myocardium. Using MRI, Wiese et al. demonstrated elevated extracellular volume in both the liver and heart of patients with cirrhosis, indicating diffuse myocardial fibrosis—an essential structural change in CCM. Similarly, Isaak et al., employing advanced contrast MRI, identified myocardial fibrosis and subclinical myocardial inflammation as contributors to both systolic and diastolic dysfunction in CCM [100,101].

The impaired compliance and relaxation reduce ventricular filling efficiency, prolonging isovolumetric relaxation time and diminishing passive early diastolic filling. As a compensatory mechanism, atrial contribution to ventricular filling increases, leading to elevated left ventricular end-diastolic pressure. The 2005 WCG criteria defined diastolic dysfunction in CCM using metrics such as the deceleration time of early ventricular filling velocity (DT > 200 ms), isovolumetric relaxation time (IVRT > 80 ms), or a reduced early-to-late filling velocity ratio (E/A ratio < 1) [11,73].

It is important to recognize that diastolic abnormalities may not be apparent at rest. Symptoms of diastolic dysfunction often emerge only during physical exertion, as left ventricular filling pressure may remain normal when the patient is at rest. However, this pressure increases with exercise due to the heart’s inability to elevate cardiac output without also increasing the filling pressure [102].

A meta-analysis by Stundiene et al., which included 16 studies, found that around 51% of cirrhotic patients exhibit diastolic dysfunction. However, these findings are constrained by the use of less specific parameters, such as the E/A ratio, and the inclusion of patients who may have ventricular relaxation impairment due to causes other than cirrhosis [103]. As highlighted by Premkumar et al. and Ruíz-del-Árbol et al., the inadequate detection of diastolic dysfunction may result in unfavorable clinical outcomes [104,105].

A study by Atroush et al. found a high prevalence of diastolic dysfunction (87.5%) among patients with end-stage liver disease, as measured by the E/e’ ratio using tissue Doppler imaging (TDI). This method was determined to be more accurate than the E/A ratio in assessing diastolic function [106].

Diastolic dysfunction is prevalent among cirrhotic patients and tends to escalate with advancing liver disease. A study demonstrated that patients with severe cirrhosis (Child–Pugh Class C) exhibited significantly higher E/Em ratios (17.0 ± 3.0) compared to those with milder forms (Child–Pugh Classes A and B: 11.5 ± 2.8), indicating impaired ventricular relaxation. Additionally, an increase in the indexed left atrial volume was observed in severe cases (34.5 ± 3.2 mL/m^2^) relative to less severe cases (30.1 ± 2.9 mL/m^2^), reflecting elevated left ventricular filling pressures. These findings suggest a direct association between diastolic dysfunction and the severity of liver cirrhosis [107].

The interplay between cardiac dysfunction and liver disease severity has significant clinical implications. Diastolic dysfunction, particularly in its more advanced stages, has been associated with adverse prognoses in patients with cirrhosis, with research demonstrating decreased survival rates in individuals with grade II diastolic dysfunction [20].

Furthermore, a 2022 study by Vetrugno et al., which included 83 recipients of orthotopic liver transplants, investigated the link between preoperative diastolic dysfunction and the risk of early allograft dysfunction. The study concluded that patients with impaired diastolic function were more likely to experience early allograft dysfunction following orthotopic liver transplantation (OLT) [108].

The importance of monitoring diastolic function in cirrhotic patients, especially those with advanced disease, lies in the potential for the early identification and management of diastolic dysfunction. Detecting and addressing diastolic dysfunction early could lead to improved clinical outcomes by preventing further complications and optimizing treatment strategies.

### 4.3. Electrophysiological Abnormalities

The most common electrophysiological disturbances in patients with cirrhosis include QT interval prolongation, impaired heart rate response (chronotropic incompetence), and electromechanical dissociation [74,109].

Mozos explored the intricate relationship between heart and liver function, highlighting how diseases in one organ can significantly impact the other [110,111]. She observed a high prevalence of arrhythmias and electrocardiographic abnormalities in individuals with liver cirrhosis. Factors contributing to the heightened arrhythmia risk included CCM, changes in cardiac ion channels, electrolyte imbalances, autonomic dysfunction, hepatorenal syndrome, metabolic disturbances, advanced age, stress, inflammatory responses, impaired drug metabolism, and the presence of other comorbid conditions. Mozos emphasized the need for vigilant monitoring of cirrhotic patients for arrhythmias, particularly when medications that prolong the QT interval are prescribed [110,111].

QT interval prolongation is a frequent finding in advanced stages of cirrhosis, affecting over 60% of cases [112]. A positive correlation has been observed between the QTc interval duration and Child–Pugh scores, with patients in Child–Pugh Class C exhibiting longer QTc intervals compared to those in Classes A and B [98]. Although the exact mechanisms underlying this abnormality are not fully understood, it is often linked to portal hypertension and porto-systemic shunting. Interestingly, this condition appears independent of the specific cause of liver disease. QT prolongation, particularly after liver transplantation, is considered a marker of poor prognosis [113,114].

This alteration in cardiac repolarization heightens the risk of torsades de pointes, ventricular arrhythmias, and sudden cardiac death. However, these severe complications are relatively uncommon in cirrhotic patients. Notably, QT prolongation during episodes of gastrointestinal bleeding correlates with lower survival rates [115,116].

Research by M. Bernardi et al. highlighted a strong association between QT prolongation, the extent of liver dysfunction, elevated plasma norepinephrine levels, and reduced survival [114]. J. Henriksen et al. suggested that beta-blockers may shorten the QT interval via a vagal mechanism [117]. Similarly, A. Zambruni et al. reported that beta-blockers decrease QT prolongation, though only in patients who initially present with prolonged QT intervals [118].

However, beta-blockers can increase the mortality risk in patients with refractory ascites and may exacerbate circulatory dysfunction. It is also essential to exercise caution with medications that can further prolong the QT interval, such as quinolones, macrolides, and agents affecting gastrointestinal motility, with careful monitoring in such cases [109,119] (Table 2).

Atrial fibrillation and flutter are irregular heart rhythms more commonly identified in patients with cirrhosis. These arrhythmias are strongly linked to conditions like arteriosclerosis, high cholesterol levels, and diabetes [120,121].

Josefsson et al. observed various supraventricular arrhythmias in cirrhotic patients, including atrial and junctional premature contractions, atrial flutter or fibrillation, and sinus tachycardia or bradycardia. In addition, pre-transplant assessments of patients with cirrhosis also revealed atrioventricular conduction abnormalities such as complete or partial right or left bundle branch blocks and other intraventricular conduction delays [122].

Inflammation may contribute to cardiac and arrhythmogenic complications in patients with MASLD. Interestingly, Zamirian et al. proposed that liver cirrhosis might have a protective effect against atrial fibrillation, despite notable metabolic abnormalities, chronic inflammation, and left atrial enlargement. Their study suggested that the low prevalence of atrial fibrillation in cirrhotic patients could result from the accumulation of anti-arrhythmic or anti-inflammatory substances that would typically be metabolized by a healthy liver. This hypothesis could also explain the onset of atrial fibrillation following liver transplantation. Further research in this area is needed to clarify this connection [123].

The low prevalence of systemic hypertension in cirrhotic patients and the use of medications such as spironolactone and beta-blockers contribute to the low incidence of atrial fibrillation. More specifically, spironolactone mitigates myocardial fibrosis in dilated atria, shortens the P-wave duration, exerts an antifibrotic effect on the ventricles, and decreases the QT interval. Beta-blockers, commonly prescribed as prophylaxis against variceal bleeding in cases of large esophageal varices, induce vasoconstriction in the splanchnic circulation, thereby increasing preload and enhancing diastolic function. However, their use in advanced decompensated cirrhosis is associated with poorer survival outcomes [119]. Beta-blockers impair atrioventricular conduction, leading to bradycardia or high-grade heart blocks [119,124,125].

Myocardial fibrosis is a known trigger for arrhythmias, and atrial interstitial fibrosis alters the electrical properties of the atria, leading to reduced excitability, increased refractoriness, and slowed or blocked conduction [126]. Angiotensin-converting enzyme (ACE) inhibitors offer protection against myocardial fibrosis, prevent cardiac remodeling, and reduce the risk of atrial fibrillation. Similarly, angiotensin II receptor blockers, which target the renin–angiotensin system, help prevent atrial remodeling. However, ACE inhibitors and other afterload-reducing drugs must be used cautiously, as they may exacerbate the vasodilatory state in cirrhotic patients [127].

Chronotropic incompetence refers to the heart’s inability to adequately increase its rate in response to physical or pharmacological stimuli. The underlying mechanisms remain unclear, but dysfunction in β-adrenergic receptor pathways has been proposed, particularly during heightened demand for β-adrenergic agonists [43,118,128,129].

Electromechanical dissociation, defined as a delay between the heart’s electrical excitation and mechanical contraction, can be assessed through systolic interval measurements or by analyzing simultaneous ECG and aortic pressure recordings. This delay tends to worsen with QT prolongation. The anomaly is thought to involve β-adrenergic receptor or post-receptor defects. However, its precise clinical significance is yet to be determined [130,131]. Table 3 summarizes studies that reported electrocardiographic abnormalities in patients with liver disease.

### 4.4. The Bidirectional Relationship Between Cirrhotic Cardiomyopathy and TIPS

CCM significantly influences the outcomes of TIPS procedures, while TIPS itself exerts notable effects on cardiac function.

A study by Cazzaniga et al. demonstrated that diastolic dysfunction, identified by an E/A ratio ≤1, was an independent predictor of mortality post-TIPS. Specifically, 60% of patients with such diastolic dysfunction succumbed within the first year post-procedure, whereas all patients with an E/A ratio >1 survived [139].

Additionally, a prospective study revealed that patients with CCM exhibited lower mean arterial pressure and left ventricular ejection fraction post-TIPS compared to those without CCM. Despite these differences, the one-year mortality rates did not differ significantly between the groups, suggesting that while CCM alters hemodynamic responses post-TIPS, it may not independently predict long-term survival [140].

The insertion of TIPS leads to immediate hemodynamic changes, including increased central blood volume and cardiac output. A study by Busk et al. observed significant elevations in cardiac output and stroke volume one week post-TIPS, accompanied by subtle yet significant changes in cardiac function parameters [141].

These findings suggest that TIPS challenges the cardiovascular system, potentially unmasking or exacerbating underlying CCM. Furthermore, the study by Cazzaniga et al. noted that while TIPS generally improved diastolic function in many patients, those with persistent diastolic dysfunction post-procedure had a poorer prognosis, highlighting the intricate interplay between TIPS and cardiac function [139].

CCM can negatively impact the hemodynamic response and survival outcomes following TIPS. Conversely, TIPS can unmask or exacerbate cardiac dysfunction in patients with underlying CCM. Therefore, thorough cardiovascular assessments are crucial when considering TIPS in patients with cirrhosis to optimize patient selection and management strategies.

### 4.5. The Impact of Cirrhotic Cardiomyopathy on Liver Transplantation Outcomes and Its Post-Transplant Evolution

CCM exerts a profound impact on perioperative and postoperative outcomes in patients undergoing OLT. The transplantation procedure introduces substantial hemodynamic stress due to fluctuations in preload and afterload, alongside the release of inflammatory mediators and vasoactive substances. Patients with CCM often exhibit a diminished capacity to adapt to these changes, leading to overt systolic and diastolic dysfunction. Notably, cardiovascular complications occur in up to 70% of liver transplant recipients, with the reperfusion phase being particularly prone to instability. Postreperfusion syndrome, characterized by a significant drop in mean arterial pressure, manifests in 12–77% of recipients [142].

Following OLT, patients with CCM are at an elevated risk for heart failure. Mortality from heart failure post-transplantation is estimated to be as high as 15%, with clinical or radiographic evidence of pulmonary edema observed in up to 56% of patients within the first postoperative week [63]. Additionally, perioperative heart failure correlates with extended intensive care unit stays, increased infection rates, and higher long-term patient and graft mortality [143].

The role of diastolic dysfunction in predicting post-transplant outcomes remains contentious. Some studies indicate that diastolic dysfunction does not correlate with adverse long-term outcomes, while others suggest a significant association with reduced survival rates over a two-year follow-up period. Moreover, higher grades of diastolic dysfunction have been linked to an increased risk of perioperative heart failure [142].

CCM often presents with prolonged rate-corrected QT intervals (QTc > 440 ms), detected in 30–60% of patients. While liver transplantation has been shown to reverse many features of CCM, including electrophysiological abnormalities, the immediate perioperative period remains critical due to potential myocardial dysfunction [63].

The impact of OLT on CCM has been a subject of clinical interest. Studies have demonstrated that certain cardiac abnormalities associated with CCM may improve following OLT. Significant reductions in left ventricular wall thickness and mass post-transplantation have been observed, suggesting a regression of myocardial hypertrophy. Additionally, improvements in diastolic function parameters, such as the E/A ratio and deceleration time, indicate a potential normalization of diastolic filling patterns. These findings imply that liver transplantation can ameliorate structural and functional cardiac alterations associated with cirrhosis [144].

Despite the potential for improvement, some studies have reported the persistence or emergence of cardiac dysfunction following OLT. For instance, cases of systolic heart failure occurring within six months post-transplantation have been identified, with pre-existing diastolic dysfunction serving as an independent predictor. The etiology of post-transplant systolic dysfunction is multifactorial, encompassing stress-induced cardiomyopathy and the unmasking of subclinical cardiac conditions. Notably, approximately half of the affected patients experienced recovery of left ventricular function, while the remainder continued to exhibit reduced ejection fractions, underscoring the heterogeneity of cardiac outcomes post-OLT [145].

The variable cardiac outcomes following liver transplantation necessitate meticulous preoperative cardiac evaluation and vigilant postoperative monitoring. Identifying patients with pre-existing diastolic dysfunction is crucial, as they are at an elevated risk for postoperative systolic heart failure. Implementing tailored perioperative management strategies can mitigate cardiac complications and enhance overall transplant success.

In conclusion, while liver transplantation holds the potential to reverse certain cardiac abnormalities associated with cirrhotic cardiomyopathy, the persistence or development of cardiac dysfunction post-transplantation remains a concern. Comprehensive cardiac assessments and individualized management are imperative to optimize outcomes for liver transplant recipients.

## 5. Conclusions

Advanced cirrhosis is associated with increased cardiac involvement, particularly in the form of left ventricular diastolic dysfunction. As the severity of liver disease progresses, the extent of cardiac dysfunction, especially in the diastolic phase, tends to worsen.

Echocardiography is a non-invasive, real-time imaging technique that provides rapid and highly accurate diagnosis of cardiac abnormalities, including those indicative of CCM. It is especially valuable for obtaining a timely diagnosis of CCM and assessing the severity of the disease, allowing for the development and initiation of appropriate clinical management strategies.

Early clinical and diagnostic assessments to detect signs of cardiomyopathy in cirrhotic patients undergoing liver transplantation are essential. These investigations are crucial to prevent the deterioration of cardiovascular hemodynamics in the post-transplant period. Early identification can help guide management strategies and improve outcomes after transplantation.

At the same time, the latent nature of cirrhotic cardiomyopathy necessitates the thorough evaluation of arrhythmia risk in cirrhotic patients. Close monitoring for arrhythmias is essential, especially when administering QT interval-prolonging medications, in the presence of electrolyte imbalances or during complications like hepatorenal syndrome. The risk of arrhythmias may persist after liver transplantation; consequently, future research should focus on the interplay between the arrhythmia risk and structural heterogeneity of the cirrhotic heart.

## Figures and Tables

**Figure 1 biomedicines-13-00812-f001:**
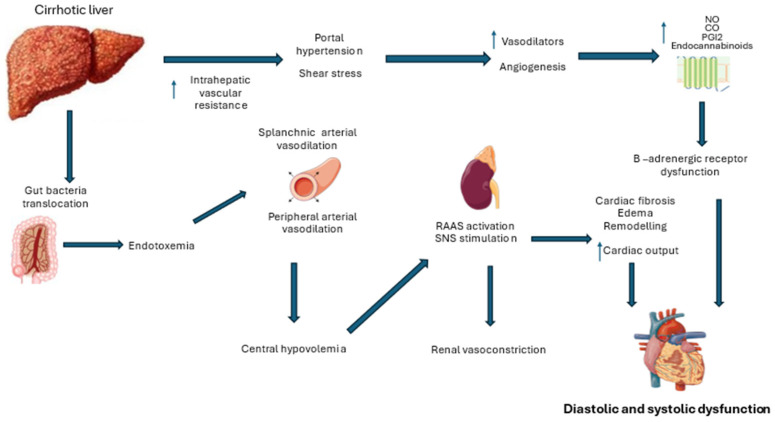
Pathophysiological mechanisms underlying cirrhotic cardiomyopathy (RAAS: renin–angiotensin–aldosterone system; SNS: sympathetic nervous system).

**Table 1 biomedicines-13-00812-t001:** The 2005 Montreal criteria and 2019 Cirrhotic Cardiomyopathy Consortium criteria for the diagnosis of CCM.

**2005 Montreal Criteria**
**Systolic Dysfunction** (*Meeting at least one*) Blunted increase in cardiac output (CO) with exercise, volume challenge, or pharmacological stimulation Resting ejection fraction of LV (LVEF) < 55% **Diastolic Dysfunction** E/A ratio < 1 (early-to-late ventricular filling ratio) Prolonged deceleration time (>200 ms) Prolonged isovolumetric relaxation time (>80 ms) **Supportive Criteria** Electrophysiological abnormalities Abnormal chronotropic response Electromechanical uncoupling Prolonged QTc interval **Structural and Biomarker Indicators** Enlarged left atrium Increased myocardial mass Elevated brain natriuretic peptide (BNP) or proBNP Increased troponin I levels
**2019 Cirrhotic Cardiomyopathy Consortium Criteria** **Diagnostic Criteria** **Systolic Dysfunction** (*Presence of any of the following*) Left ventricular ejection fraction (LVEF) ≤ 50% Absolute global longitudinal strain (GLS) < 18% **Diastolic Dysfunction** (*Presence of at least three of the following*) Septal e’ velocity < 7 cm/s E/e’ ratio ≥ 15 Left atrial volume index (LAVI) > 34 mL/m^2^ Tricuspid regurgitation (TR) velocity > 2.8 m/s

BNP: brain natriuretic peptide; CO: cardiac output; GLS: global longitudinal strain; LA: left atrial; LAVi: left atrial volume index; LV: left ventricular; LVEF: left ventricular ejection fraction; TR: tricuspid regurgitation.

**Table 2 biomedicines-13-00812-t002:** Causes of QT prolongation in liver cirrhosis.

Autonomic neuropathy
The stage of liver disease and the existence of portal hypertension
Serum markers (electrolytes, creatinine, and biochemical makers)
Volume overload (dimensions and function of the left ventricle) and left ventricular hypertrophy
Coronary heart disease (evaluation of risk factors)
Coexisting stressful events (bleeding)
Drugs (macrolides and quinolones)

**Table 3 biomedicines-13-00812-t003:** Summary of studies concerning the electrocardiographic abnormalities and the arrhythmic burden in patients with liver disease.

Author	Number of Patients	Main Findings
F Gundling et al. [121]	293	The prevalence of atrial fibrillation was increased. Rhythm disorders were linked with risk factors. Diuretic therapy and electrolyte disturbances lead to cardiac arrhythmia.
A Zambruni et al. [132]	100	Bazett’s correction should be avoided in cirrhotic patients. Fridericia’s formula can be used.
Toma et al. [133]	117	Prolonged QT intervals in advanced liver disease. QRS amplitude was lower in decompensates cirrhosis.
Pourafkari et al. [134]	69	Electrocardiographic abnormalities are frequently observed in patients with cirrhosis, irrespective of the severity of the disease. Low-voltage QRS complexes may be associated with anatomical changes and the presence of ascites in these individuals.
Jahangiri et al. [135]	425	ECG changes in prolonged QT and early transitional zones were related to the severity of cirrhosis.
Josefsson et al. [122]	234	Electrocardiographic abnormalities are frequently observed in patients with cirrhosis before liver transplantation and are linked to cardiovascular risk factors as well as the severity and underlying cause of cirrhosis. Cardiac events following transplantation occur more often in these patients compared to the general population.
Singh et al. [136]	50	QTc prolongation and low-voltage QRS complexes are closely associated with the severity of liver cirrhosis, as reflected by their correlation with the Child–Turcotte–Pugh (CTP) and model for end-stage liver Disease (MELD) scores. These electrocardiographic abnormalities are more frequently observed in patients with complications of decompensated cirrhosis, such as ascites and hepatic encephalopathy.
Lu et al. [120]	135	Age and the presence of ascites have been identified as significant risk factors for the development of atrial arrhythmias in patients with liver cirrhosis.
Dimala et al. [137]	9,612,601 (meta-analysis)	Metabolic-associated steatotic liver disease (MASLD) is linked to various electrocardiographic abnormalities, which may serve as early indicators of cardiac involvement. These findings underscore the multisystemic nature of MASLD. Incorporating these specific ECG changes into screening and management protocols could enhance cardiac risk stratification in patients with MASLD.
Huang et al. [138]	1727	The prevalence and incidence of atrial fibrillation (AF) are elevated in patients with liver disease. The severity of liver disease, as assessed by the model for end-stage liver disease (MELD) score, is a significant predictor of new-onset AF.

## Data Availability

Not applicable.

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
