# Peer review of "Cardiac Remodeling and Arrhythmic Burden in Pre-Transplant Cirrhotic Patients: Pathophysiological Mechanisms and Management Strategies"

_biomedicines, 2025, doi:10.3390/biomedicines13040812_

Round 1
Reviewer 1 Report
Comments and Suggestions for Authors
This review is well thought out, well written in a logical manner, and covers the literature extensively. The authors cover the current literature, including other reviews and meta-analyses to provide a comprehensive review. Further, the authors are not overly cited in their own review. I have no major concerns regarding this review and only one minor concern about the title. Most of the review discusses the left ventricle, however there is some attention given to atrial remodeling and fibrillation and this could be reflected in the title.
Author Response
Comment: This review is well thought out, well written in a logical manner, and covers the literature extensively. The authors cover the current literature, including other reviews and meta-analyses to provide a comprehensive review. Further, the authors are not overly cited in their own review. I have no major concerns regarding this review and only one minor concern about the title. Most of the review discusses the left ventricle, however there is some attention given to atrial remodeling and fibrillation and this could be reflected in the title.
Response:
Dear Reviewer,
Thank you for your thoughtful and positive feedback. We greatly appreciate your recognition of the comprehensiveness and logical structure of our review. In response to your valuable suggestion regarding the title, we have revised it to more accurately capture the broader scope of the review, particularly with regard to highlighting the arrhythmic load in patients with advanced cirrhosis as it manifests through various types of arrhythmias. The updated title is now:
‘Cardiac Remodelling and Arrhythmic Burden in Pre-Transplant Cirrhotic Patients: Pathophysiological Mechanisms and Management Strategies’
Reviewer 2 Report
Comments and Suggestions for Authors
In this review, the authors described the interplay between liver cirrhosis and cardiac dysfunction, emphasizing the pathophysiology, diagnostic criteria, and management of cirrhotic cardiomyopathy.
Comments on improving the quality of the manuscript.
1. Please provide up-to-date data on systemic hemodynamic disorders contributing to the development of cirrhotic cardiomyopathy (see, for example, DOI: 10.1016/j.jhep.2021.01.002).
2. I strongly recommend that the authors revise the section "3.3. Causes that lead to cardiac dysfunction in cirrhotic patients". It is necessary to provide current information on the pathophysiological mechanisms of cardiovascular disorders in non-alcoholic fatty liver disease/metabolic dysfunction–associated steatotic liver disease
(see, for example, DOI: 10.22037/ghfbb.v15i3.2549).
3. I suggest to describe in more detail the alterations to cardiac morphology and function in cirrhotic cardiomyopathy, their relationship with the severity of liver cirrhosis.
4. Please describe the effect of cirrhotic cardiomyopathy on the outcome of TIPS and the impact of TIPS on cirrhotic cardiomyopathy.
5. Please describe the effect of cirrhotic cardiomyopathy on the outcome of liver transplantation.
6. What happens to cirrhotic cardiomyopathy after liver transplantation?
7. I also advise to revise the manuscript title.
Author Response
Dear Reviewer,
Thank you for your valuable feedback and thoughtful comments on our manuscript entitled [Left ventricular remodelling and arrhythmias in pre-transplantation cirrhotic patients]. We greatly appreciate your time and effort in reviewing our work. Below, we address each of your comments and provide the corresponding revisions.
Comment 1: Please provide up-to-date data on systemic hemodynamic disorders contributing to the development of cirrhotic cardiomyopathy (see, for example, DOI: 10.1016/j.jhep.2021.01.002).
Response: We have made an effort to include more recent data regarding systemic hemodynamic disorders contributing to the development of cirrhotic cardiomyopathy, strengthening Section 3.1.
Comment 2: I strongly recommend that the authors revise the section "3.3. Causes that lead to cardiac dysfunction in cirrhotic patients". It is necessary to provide current information on the pathophysiological mechanisms of cardiovascular disorders in non-alcoholic fatty liver disease/metabolic dysfunction–associated steatotic liver disease
(see, for example, DOI: 10.22037/ghfbb.v15i3.2549).
Response: Thank you for your suggestion. We have revised Section 3.3 to include current information on the pathophysiological mechanisms of cardiovascular disorders in metabolic dysfunction–associated steatotic liver disease (MASLD). We have updated this section with the most recent findings to provide a comprehensive understanding of the cardiovascular implications of these conditions. Relevant references have been added to support this updated discussion.
Comment 3: I suggest to describe in more detail the alterations to cardiac morphology and function in cirrhotic cardiomyopathy, their relationship with the severity of liver cirrhosis.
Response: We have carefully followed your recommendation to include studies that explore the association between cardiac dysfunction, structural changes, and the progression of cirrhosis. We have strengthened our manuscript in Section 4.1,4.2 and 4.3 incorporating additional details and references to further enhance the discussion.
Comment 4: Please describe the effect of cirrhotic cardiomyopathy on the outcome of TIPS and the impact of TIPS on cirrhotic cardiomyopathy.
Response: Following your suggestion, we aimed to highlight the bidirectional relationship between cirrhotic cardiomyopathy and TIPS. We have expanded our manuscript to address the effect of cirrhotic cardiomyopathy (CCM) on the outcome of transjugular intrahepatic portosystemic shunt (TIPS) and vice versa. CCM can complicate TIPS outcomes by increasing the risk of cardiovascular instability and post-TIPS complications. On the other hand, while TIPS can reduce portal hypertension and improve liver function, it may also affect cardiac function by altering hemodynamics, potentially worsening CCM in some cases. We have elaborated on these points in the manuscript and added relevant references to support our discussion (section 4.4).
Comment 5 and Comment 6: Please describe the effect of cirrhotic cardiomyopathy on the outcome of liver transplantation. What happens to cirrhotic cardiomyopathy after liver transplantation?
Response: Thank you for your insightful comments. We have now included a detailed discussion regarding the impact of cirrhotic cardiomyopathy on liver transplantation outcomes in our manuscript. Cirrhotic cardiomyopathy has been shown to influence both the perioperative and long-term outcomes of liver transplantation, as it may exacerbate cardiovascular complications such as impaired myocardial contractility and hemodynamic instability. After liver transplantation, cirrhotic cardiomyopathy may improve to some extent due to the resolution of liver dysfunction; however, it is not always completely reversible. We have elaborated on these points by creating a new section in the manuscript and added relevant references to support this discussion (section 4.5).
Comment 7: I also advise to revise the manuscript title
Response: We appreciate your suggestion. After careful consideration, we have revised the manuscript title to more accurately reflect the content and focus of the study. The new title is:
Cardiac Remodelling and Arrhythmic Burden in Pre-Transplant Cirrhotic Patients: Pathophysiological Mechanisms and Management Strategies
Once again, thank you for your constructive suggestions, which have significantly contributed to improving the quality of our manuscript.
Reviewer 3 Report
Comments and Suggestions for Authors
Cardiac function evaluation is important in the management of patients with liver disease, considering the impact of liver conditions on heart function and the impact of cardiac function on the outcome, particularly in patients undergoing liver transplantation. In this setting, cirrhotic cardiomyopathy may be a neglected entity that requires an adequate highlight to optimize the treatment strategy and outcome of patients with severe liver disease.
Comments on the Quality of English LanguageThe English Language requires minor revision.
Author Response
Comment: Cardiac function evaluation is important in the management of patients with liver disease, considering the impact of liver conditions on heart function and the impact of cardiac function on the outcome, particularly in patients undergoing liver transplantation. In this setting, cirrhotic cardiomyopathy may be a neglected entity that requires an adequate highlight to optimize the treatment strategy and outcome of patients with severe liver disease.
Comments on the Quality of English Language
The English Language requires minor revision
Response:
Dear Reviewer
Thank you for your insightful comment. We fully agree that cardiac function evaluation plays a critical role in the management of patients with liver disease, particularly in the context of liver transplantation. We have made sure to emphasize the importance of recognizing cirrhotic cardiomyopathy, as it can significantly influence both treatment strategies and patient outcomes. We have enhanced the discussion in the manuscript to highlight the role of cirrhotic cardiomyopathy more thoroughly, ensuring that its implications for the management of patients with severe liver disease are adequately addressed. Minor grammatical and stylistic refinements have been made to enhance readability and ensure consistency.
e.g.
Abstract
Structural changes, including increased left ventricular mass, myocardial fibrosis, and ion channel dysfunction, further compromise cardiac function. Vasodilation in the splanchnic circulation reduces peripheral resistance, inducing compensatory tachycardia, while renin-angiotensin-aldosterone system (RAAS) activation promotes fluid retention and cardiac preload.
After the revision.
Structural changes, such as increased left ventricular mass, myocardial fibrosis, and ion channel dysfunction, further impair cardiac function. Vasodilation in the splanchnic circulation reduces peripheral resistance, triggering compensatory tachycardia, while activation of the renin-angiotensin-aldosterone system (RAAS) promotes fluid retention and increases cardiac preload.
Section 3.3 (line 239)
- Alcohol – induced dilated cardiomyopathy can develop even in the absence of cirrhosis. However, with up to a million people affected by alcohol misuse disorder, the condition is likely underdiagnosed.
After the revision.
- Alcohol-induced dilated cardiomyopathy can occur even in the absence of cirrhosis. However, with alcohol use disorder affecting up to a million individuals, the true prevalence of this condition is likely underestimated and underdiagnosed.
Section 4.1(line 355)
Research indicating a reduction in LVEF following liver transplantation—linked to a rapid rise in afterload and the resolution of hyperdynamic circulation—suggests the presence of an inherent contractile defect in CCM patients.
After the revision.
Evidence indicating a post-transplant decline in LVEF, attributed to a rapid increase in afterload and the resolution of hyperdynamic circulation, suggests an underlying contractile dysfunction inherent to CCM.
Round 2
Reviewer 2 Report
Comments and Suggestions for Authors
No Comments
Comments on the Quality of English LanguageNo Comments